# Deep Non-line-of-sight Imaging from Under-scanning Measurements

**Yue Li**  **Yueyi Zhang**[*]  **Juntian Ye**  **Feihu Xu**  **Zhiwei Xiong**
University of Science and Technology of China
{yueli65,jt141884}@mail.ustc.edu.cn
{zhyuey,feihuxu,zwxiong}@ustc.edu.cn

## Abstract

Active confocal non-line-of-sight (NLOS) imaging has successfully enabled seeing around corners relying on high-quality transient measurements. However, acquiring spatial-dense transient measurement is time-consuming, raising the question of how to reconstruct satisfactory results from under-scanning measurements (USM). The existing solutions, involving the traditional algorithms, however, are hindered by unsatisfactory results or long computing times. To this end, we propose the first deep-learning-based approach to NLOS imaging from USM. Our proposed end-to-end network is composed of two main components: the transient recovery network (TRN) and the volume reconstruction network (VRN). Specifically, TRN takes the under-scanning measurements as input, utilizes a multiple kernel feature extraction module and a multiple feature fusion module, and outputs sufficient-scanning measurements at the high-spatial resolution. Afterward, VRN incorporates the linear physics prior of the light-path transport model and reconstructs the hidden volume representation. Besides, we introduce regularized constraints that enhance the perception of more local details while suppressing smoothing effects. The proposed method achieves superior performance on both synthetic data and public real-world data, as demonstrated by extensive experimental results with different under-scanning grids. Moreover, the proposed method delivers impressive robustness at an extremely low scanning grid (i.e., $8\times8$) and offers high-speed inference (i.e., 50 times faster than the existing iterative solution). The code is available at https://github.com/Depth2World/Under-scanning_NLOS.

## 1 Introduction

Non-line-of-sight (NLOS) imaging technology permits the detection of objects located behind occluders, and has numerous real-world applications, such as autonomous driving, remote sensing, and disaster rescue [1, 2, 3, 4, 5, 6, 7, 8, 9]. The widely-known time-of-flight confocal imaging system, composed of a pulse laser, a Single Photon Avalanche Diode (SPAD), and a Time-Correlated Single Photon Counting sensor (TCSPC), is favored for its ability to capture high-quality transient measurements [10]. As illustrated in Fig. 1, the picosecond pulse laser illuminates the relay wall at a certain location. The light scatters at the target object behind the wall and then re-scatters to the relay wall, from which the photons are captured by the SPAD along with TCSPC, forming a histogram. The transient measurement records the histograms of all raster-scanned locations and can be used to derive the hidden object volume by the well-studied reconstruction algorithms such as those developed in [5, 11, 12, 13, 14, 15]. However, the total acquisition time, which is directly proportional to the number of scanning points and the exposure time of each scanning point, hinders the real-time NLOS imaging process.

---

[*]Corresponding author

37th Conference on Neural Information Processing Systems (NeurIPS 2023).

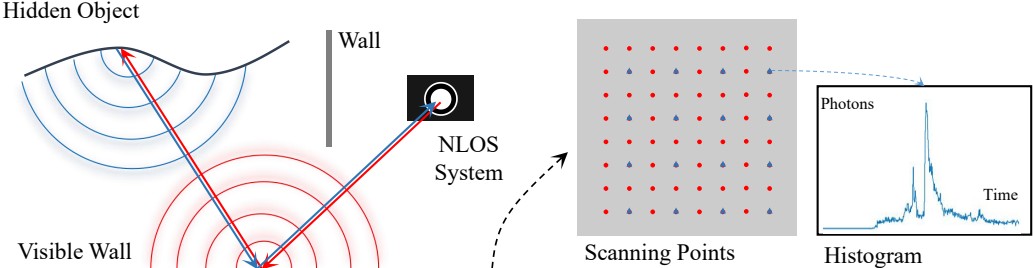

Figure 1: Overview of the active confocal NLOS imaging. The red circles denote a sufficient-scanning grid at the high-spatial resolution, and the blue triangles denote an under-scanning grid.

The time-consuming dilemma raises the question of how to reconstruct the hidden volume from the under-scanning measurement (USM) without compromising the imaging quality. As shown in Fig. 1, the spherical waves are scattered back by the illuminated object, which indicates that the temporal histogram from each scanning point contains all the geometry information of the hidden object. As a result, it is feasible to recover the high-spatial-resolution hidden volume from USM. Previous studies [16, 17] explored NLOS imaging from USM using circular scanning or the compressed sensing algorithm [18]. These solutions either produce unsatisfactory outcomes or require lengthy numerical iterative computations, leaving a large room for improvement.

In this paper, we propose a new approach to NLOS reconstruction from USM, pioneering the utilization of deep learning to achieve superior quality and swift inference. Specifically, our proposed network consists of two main components: the transient recovery network (TRN) and the volume reconstruction network (VRN). Given the under-scanning transient measurement, a multiple kernel feature extraction module in TRN generates the features at different kernel scales to obtain the pyramid receptive field. Then, the features are concatenated and aggregated by a multiple feature fusion module to produce the high-spatial-resolution transient measurement which undergoes VRN subsequently. The VRN consists of a feature extraction module, a feature propagation module, a volume refinement module, and a projection reconstruction module. Inspired by [19, 5, 3, 6], we integrate the linear physics prior [12] into the feature propagation module. Additionally, we introduce the local similarity and total variation losses to regularize our network. The local similarity loss assists with perceiving more local details, and the total variation loss aids in suppressing smoothing effects. Our proposed network has an end-to-end architecture that necessitates only one forward pass for reconstruction. The high-fidelity reconstruction in an extremely low scanning grid (i.e., $8 \times 8$) and the high-speed inference (i.e., 50 times faster than the existing iterative solution) demonstrate the superiority of our proposed method. The contributions are summarized as follows:

- We study the challenging task of NLOS imaging from USM and develop an end-to-end network for high-quality and fast inference, which is the first of its kind.

- We exploit the multiple kernel mechanism in the transient recovery network to adapt transient measurements with different under-scanning grids.

- Our proposed method achieves superior performance on both synthetic data and public real-world data, as demonstrated by extensive experimental results.

## 2 Related Work

### 2.1 NLOS Imaging Algorithms

In recent years, transient-based NLOS imaging has garnered significant attention in the research community. Despite the apparent progress made by direct reconstruction methods [11, 12, 13, 14, 15, 20, 21, 22, 23], these methods remain sensitive to exposure noise, data quality, and variations in regions. Concurrently, iterative methods [2, 24, 25, 26, 27] can effectively reduce imaging noise but suffer considerably from lengthy computation time. In contrast, deep-learning-based techniques [28, 5, 3, 29] offer a balance between the imaging quality and inference time, making them an attractive proposition for NLOS reconstruction. To this end, we approach NLOS reconstruction

from a deep-learning perspective, leveraging its advantages of efficient inference and robustness to noisy data.

## 2.2 NLOS Imaging From USM

The experiments conducted in [12, 17] indicated that the exposure time per scanning point is more crucial than the number of scanning points for imaging quality. Subsampling the scanning points can be a viable trade-off for reducing the total acquisition time. C$^2$NLOS [16] advocated for circular scanning methodology which improved the transient capturing efficiency, but the corresponding reconstruction quality left much to be desired. CSA [17] investigated a compressed sensing algorithm called SPITAL [18] for NLOS imaging from USM at a grid shape and demonstrated impressive results at a low scanning ratio, however, its iterative computation required an extended time. In contrast, our proposed method leverages deep-learning technology for NLOS imaging from USM, and necessitates a single forward pass for inference. As a concurrent work, Wang *et al.* [30] utilized the learning-based method and focused on recovering the high-spatial-resolution transient measurement, rather than NLOS reconstruction from USM.

## 3  Problem Formulation

The travel time of the direct photons between the visible wall and the NLOS imaging system is disregarded for simplicity. The transient measurement, which solely describes the indirect photon bounces, is constructed via a 3D spatial-temporal (2D spatial and 1D temporal) representation. Similar to the previous approaches [12, 16, 22], we assume that light scatters just once at the hidden object in an isotropic manner, without any occlusions in the object. Given the illuminated point at t = 0, the object $\mathbf{o}$ and the scanning point $\mathbf{s}$ which is located at the same position as the illuminated point, the active confocal NLOS imaging observation model can be expressed as follows:

$$\tau(\mathbf{s}, t) = \iiint_\Omega \rho(\mathbf{o}) \frac{\delta\left(2||\mathbf{s} - \mathbf{o}|| - c \cdot t\right)}{||\mathbf{s} - \mathbf{o}||^4} d\Omega, \tag{1}$$

where $\tau$ denotes the transient measurement, $\Omega$ denotes the spherical waves of the scattered light, $\rho$ represents the volume of the albedo of the hidden object surface, $\delta$ models the light propagation function, $||\mathbf{s} - \mathbf{o}||^4$ is the quartic radiance falloff of light transport between the relay wall and the object, and $c$ is the speed of the light. As discussed in [12, 16], Eq. 1 can be derived to a straightforward 3D convolution form via variables changing. After discretized, the formation model can be regarded as a linear model:

$$\boldsymbol{\rho} = \boldsymbol{A}^{-1}\boldsymbol{\tau}, \tag{2}$$

where $\boldsymbol{A}$ refers to the light transport matrix, the vectorized $\boldsymbol{\rho} \in \mathbb{R}^{h_x w_y n_z}$, the vectorized $\boldsymbol{\tau} \in \mathbb{R}^{h_x w_y n_t}$, $h_x$ and $w_y$ are the number of the scanning points along the $x$ and $y$ axis, $n_t$ and $n_z$ are the length of the discretized histogram bins along time $t$ axis and depth bins along depth $z$ axis. The determination of $\boldsymbol{\rho}$ represents an inverse problem, involving 3D deconvolution procedures.

For NLOS imaging from USM, define the under-scanning transient measurement $\boldsymbol{\tau}_u \in \mathbb{R}^{h_x w_y n_t}$ and the corresponding hidden volume $\boldsymbol{\rho}_u \in \mathbb{R}^{h_x w_y n_z}$, the sufficient-scanning transient measurement $\boldsymbol{\tau}_s \in \mathbb{R}^{H_x W_y n_t}$ and the corresponding hidden volume $\boldsymbol{\rho}_s \in \mathbb{R}^{H_x W_y n_z}$, where $H_x > h_x$ and $W_y > w_y$. The NLOS imaging from USM is transformed into the following optimization problem:

$$\hat{\theta}_\mathcal{T}, \hat{\theta}_\mathcal{V} = \underset{\theta_\mathcal{T}, \theta_\mathcal{V}}{\arg\min}(\mathcal{L}(\boldsymbol{\tau}_u^\uparrow, \boldsymbol{\tau}_s) + \mathcal{L}(\boldsymbol{\rho}_u^\uparrow, \boldsymbol{\rho}_s) + \lambda\phi(\theta_\mathcal{T}, \theta_\mathcal{V})), \tag{3}$$

where $\boldsymbol{\tau}_u^\uparrow$ and $\boldsymbol{\rho}_u^\uparrow$ are the predicted high-spatial-resolution transient data and hidden volume through the algorithms $\theta_\mathcal{T}$ and $\theta_\mathcal{V}$ respectively, $\mathcal{L}$ represents the loss function between the generated and ground-truth one, $\phi$ is the regularization term, and $\lambda$ is a parameter to weigh the regularization term.

## 4  Proposed Method

In this section, we present an end-to-end deep network for NLOS imaging from USM. The network takes in under-scanning transient measurement and finally outputs the hidden volume and the intensity image at a high-spatial resolution. The overall network comprises two stages: the Transient Recovery Network (TRN) and the Volume Reconstruction Network (VRN). In the first stage, the TRN recovers

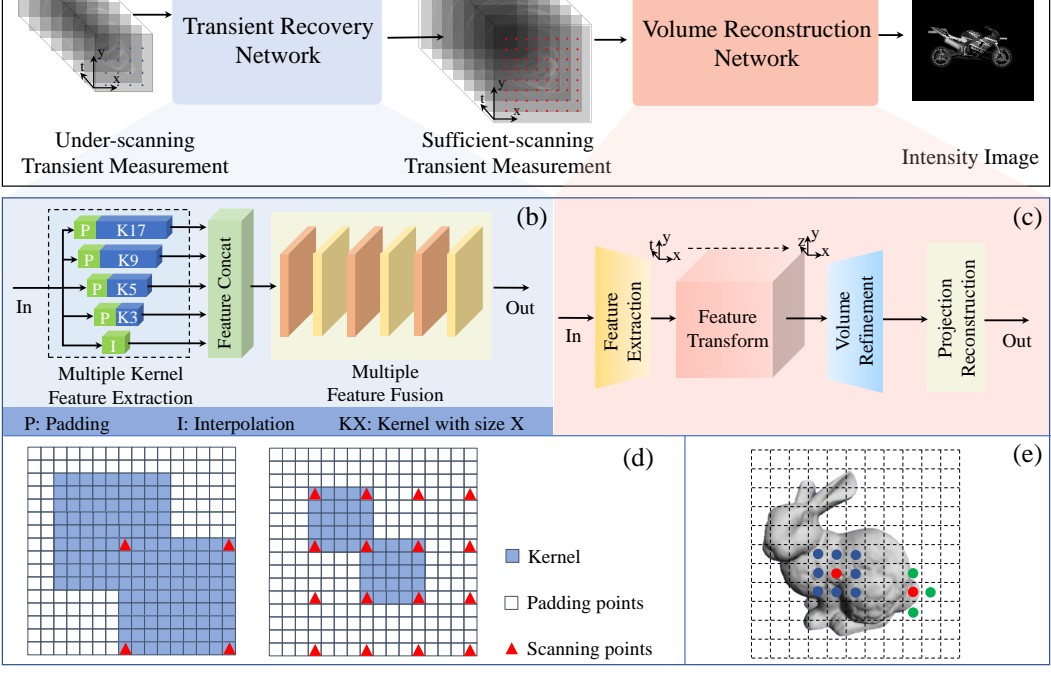

Figure 2: (a) The pipeline of the proposed network. The network consists of a transient recovery network (TRN) and a volume reconstruction network (VRN). (b) An overview of TRN. The blue rectangular blocks denote 3D residual blocks with corresponding kernel sizes. The orange and yellow rectangular blocks represent the 3D convolutions and 3D residual blocks, respectively. (c) An overview of VRN. (d) A diagram for the multiple kernel feature extraction. (e) A diagram for the regularized loss functions. The points on the objects always have similar values with the neighbors, while points on the edges have different values from the outside.

the under-scanning transient data to the desired high-spatial resolution. In the second stage, the VRN reconstructs the hidden volume and then generates the intensity image, as illustrated in Fig. 2(a).

## 4.1 Transient Recovery Network

Histograms captured from distinct scanning points all contain the complete hidden geometry information. However, each histogram has unique characteristics due to the differences in traveling time. Consequently, the data from even the adjacent scanning points can not be identical, but similar. To explore this information and restore the transient measurement at the high-spatial resolution, we design the transient recovery network (TRN) to recover sufficient-scanning transient measurement.

TRN consists of a Multiple Kernel Feature Extraction module (MKFE) and a Multiple Feature Fusion module (MFF), as illustrated in Fig. 2(b). The MKFE module comprises four residual blocks with different convolution kernels and an interpolation operator. Each residual block consists of two 3D convolutions, two Leaky ReLU activation functions, and one residual connection. The kernel scales in the residual blocks of the MKFE are arranged from large to small to perceive the entire geometry, therefore providing a sizeable receptive field. As illustrated in Fig. 2(d), the scanning points in the padded scanning grid are sparse and distant from each other. To exploit the feature connection between the scanning points in one convolution operation directly, the size of the convolution kernel is set to cover the farthest scanning point distance. The large kernel perceives the distant regions, and the small kernel focuses on the local areas. Various under-scanning grids require different appropriate kernel sizes. For a more general architecture, we further implement a multiple kernel mechanism for TRN to adapt the different under-scanning transient measurements. The MFF module alternately involves three 3D convolutions and three 3D residual blocks with fixed kernels.

Given the under-scanning transient measurement $\tau_u$, we first pad the under-scanning transient measurement to reach the desired high-spatial resolution. All histograms, except the initial position,

are zeroed. Next, we pass the padded transient data through four residual blocks to obtain fine features. Then, we utilize the nearest interpolation for the input under-scanning transient measurement to get coarse features at the desired resolution, which facilitates stable training and fast convergence. Afterward, the coarse and fine features are concatenated along the channel dimension. Then the combined features are supplied to the fusion recovery decoder to generate the sufficient-scanning transient measurement $\boldsymbol{\tau}_s$.

## 4.2 Volume Reconstruction Network

For the hidden volume reconstruction network (VRN), we borrow the idea from the previous works [5, 3] which incorporated the physics prior to feature propagation. Inspired by LFE [5], VRN is composed of a feature extraction module, a feature transform module, a volume refinement module, and a projection reconstruction module, as shown in Fig. 2(c). Different from LFE [5], we utilize the physical model-based approach called LCT [12] to build the feature transform module. LCT [12] adopts deconvolution operation and conducts efficient computation. Moreover, we design a volume refinement module to enhance the 3D features for high fidelity. The module is constructed with three 3D convolutions and three interlaced residual layers. Each of these residual layers is composed of two 3D convolutions, followed by a ReLU operation and a residual connection. This module has been purposefully designed to proficiently accentuate essential features while amplifying the 3D volume. The objective is to enhance fidelity, thereby avoiding a direct projection of the 3D volume onto 2D planes which could potentially result in losing crucial inherent features.

Given the sufficient-scanning transient measurement $\boldsymbol{\tau}_s$ which is recovered by TRN, we first extract the intrinsic spatial-temporal features by the feature extraction module. Then the extracted features are supplied to the feature transform module, where the 3D spatial-temporal features are transformed into 3D spatial features. Afterward, the 3D spatial features proceed to the volume refinement module to produce the reconstructed 3D volume $\boldsymbol{\rho}$. For further optimizing the output intensity images, the 3D volume representations are mapped to the 2D domain using maximum intensity projection along the depth axis, and then the projected 2D features are convolved to generate the final intensity images $\boldsymbol{I}$.

## 4.3 Regularized Training

Our proposed network is trained in an end-to-end supervised manner. The overall loss function consists of four components: the measurement loss $\mathcal{L}_{mea}$, the intensity loss $\mathcal{L}_{int}$, the local similarity loss $\mathcal{L}_{ls}$ and the total variation loss $\mathcal{L}_{tv}$. The total loss is formulated as follows:

$$\mathcal{L}_{total} = \mathcal{L}_{mea} + \lambda_1 \cdot \mathcal{L}_{int} + \lambda_2 \cdot \mathcal{L}_{ls} + \lambda_3 \cdot \mathcal{L}_{tv} , \tag{4}$$

where $\lambda_1$, $\lambda_2$, and $\lambda_3$ are the hyper-parameters that weigh the corresponding loss function.

To achieve end-to-end training, the L1 norms for the outputs from TRN and VRN are computed:

$$\mathcal{L}_{mea} = ||\boldsymbol{\tau}_s - \boldsymbol{\tau}_s^{gt}||_1, \quad \mathcal{L}_{int} = ||\boldsymbol{I} - \boldsymbol{I}^{gt}||_1, \tag{5}$$

where $\boldsymbol{\tau}_s$ and $\boldsymbol{\tau}_s^{gt}$ correspond to the predicted sufficient-scanning and the ground-truth transient measurements, $\boldsymbol{I}$ and $\boldsymbol{I}^{gt}$ refer to the predicted and ground-truth intensity images of the objects hidden from view. The efficacy of $\mathcal{L}_{mea}$ is showcased in Sec 5.5.

Under conditions of fewer scanning points, two problems are encountered in deep learning methodology. Firstly, the reduction of transient measurements causes difficulties in volume reconstruction, making small or distant objects mostly unrecoverable or partially recovered. Secondly, effective signal reduction or signal sparsity increase leads to network overfitting to the simulation data, thereby impeding accurate real-world testing. To overcome these challenges, we incorporate regularized constraints during the training process. Firstly, we leverage the local similarity of the natural volume since the points on the hidden object are analogous to their neighbors. We employ a Gaussian window to weigh the neighbors of the points, as [31, 32, 25]. The local similarity loss is formulated as follows:

$$\mathcal{L}_{ls} = \sum_x \sum_y \sum_z ||\boldsymbol{\rho}(x,y,z) - \hat{\boldsymbol{\rho}}(x,y,z,k) \cdot W||_1, \tag{6}$$

where $\boldsymbol{\rho}(x,y,z)$ indicates the volume at position $(x,y,z)$, $\hat{\boldsymbol{\rho}}(x,y,z,k)$ represents the volume block centered at $(x,y,z)$ with size $k$, $W$ refers to the Gaussian window with size $k$. As shown in Fig. 2(e), when the local similarity loss function acts on the points located at the objects, it brings out more local details. On the contrary, when it acts on the boundary points, it blurs and smoothens the edges. Secondly, to suppress background noise and sharpen the edge boundary, we introduce the 3D total

variation loss function, formulated as follows:

$$\mathcal{L}_{tv} = \sum_x \sum_y \sum_z (||\boldsymbol{\rho}(x+1,y,z) - \boldsymbol{\rho}(x,y,z)||_1 + ||\boldsymbol{\rho}(x,y+1,z) - \boldsymbol{\rho}(x,y,z)||_1$$
$$+ ||\boldsymbol{\rho}(x,y,z+1) - \boldsymbol{\rho}(x,y,z)||_1). \tag{7}$$

The effectiveness of the regularized loss functions $\mathcal{L}_{ls}$ and $\mathcal{L}_{tv}$ is demonstrated in Sec 5.5.

# 5 Experiments

## 5.1 Datesets and Evaluation Metrics

We evaluate our method on both synthetic and real-world datasets. For the synthetic dataset, we simulate the data using a transient renderer provided by [5], rendering 277 motorbikes from the ShapeNet dataset [33]. The synthetic dataset is composed of 2712 training samples and 291 testing samples. The scanning grid is with a size of 128×128 and the time resolution is 512 with a bin width of 33ps. For the real-world scenes, we adopt the real-world dataset provided by [14] which contains diverse objects. We conduct preprocessing to discard the direct bounce data and resize the data to 128×128×512. The total acquisition time of the real-world transient data is about 45min. We directly extract the scanning points from the initial transient data to simulate the under-scanning process. A 32×32, 16×16, and 8×8 scanning grid represents the total acquisition time of 169s, 42s, and 11s respectively. NLOS imaging from USM is able to expedite the data capturing by orders of magnitude. Besides, we also adopt the real-world data from [29] to further validate the generalization capability of our model.

For the intensity recovery results, we evaluate the performance with three metrics: Peak Signal-to-Noise Ratio (PSNR), Structural SIMilarity (SSIM), and ACCuracy (ACC). We utilize ACC to determine the foreground and background classification accuracy.

## 5.2 Implementation and Baseline Details

The proposed network is implemented in PyTorch. The optimizer is AdamW [34] with a learning rate of $1 \times 10^{-4}$ and a weight decay of $1 \times 10^{-4}$. The models are trained on the synthetic dataset for 50 epochs with a batch size of 4, and directly tested on the real-world dataset, setting the high-spatial resolution to 128×128. For the loss function, the hyperparameters $\lambda_1$, $\lambda_2$, and $\lambda_3$ is 1, $1 \times 10^{-5}$, and $1 \times 10^{-6}$. All the experiments are conducted on a workstation equipped with 4 NVIDIA GeForce A100 GPUs.

Our baseline methods include four traditional methods (FBP [11], LCT [12], FK [14], RSD [15]), three deep-learning-based methods (UNet [28], NeTF [35], LFE [5]), and one iterative method (CSA [17]). All methods, except CSA [17] perform non-iterative reconstruction. We find the performance of these non-iterative methods, which upsample the input before reconstruction, is better than that of the methods which reconstruct the volume before upsampling the output. Thus, for these non-iterative baselines, we interpolate the input under-scanning data to the desired high-spatial resolution using the nearest interpolation approach, as [17]. CSA [17] and NeTF [35] are fed with the padded transient data (at the same spatial resolution as the desired transient data) and under-scanning transient data, respectively.

## 5.3 Experimental Results

### 5.3.1 Transient Recovery

To validate the effectiveness of TRN, we compare the qualitative histograms of the recovered, direct interpolated, and raw transient measurement which is from the real-world scene (Teaser [14]) at certain scanning points. We select the transient data at three low scanning girds: 32×32, 16×16, and 8×8. As illustrated in Fig. 3, our predicted results show greater similarity to the ground truth in terms of curve trend and peak size. Specifically, under the 8×8 scanning condition, the interpolated histogram consistently generates larger outliers, while our method delivers robust performance, indicating the superiority of our proposed TRN and paving the way for subsequent volume reconstruction.

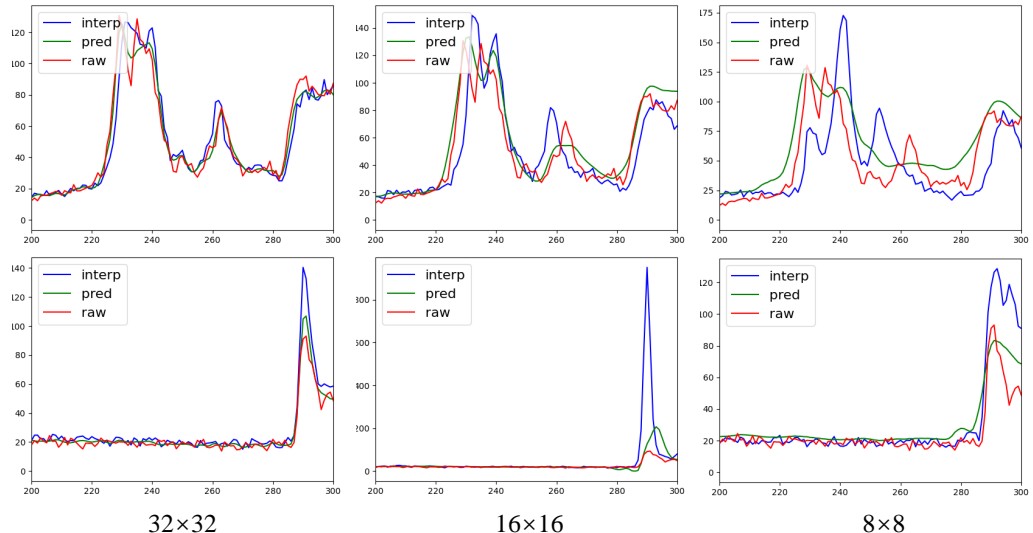

|  | | 32×32 | | 16×16 | | 8×8 |

Figure 3: Comparisons of the histograms from the recovered and the interpolated transient measurement. The rows show the histograms from different recovered transient measurements with the same scanning position. The green, blue, and red lines represent the interpolated, recovered, and ground-truth histograms. The data is from the real-world scene (Teaser in [14]).

Table 1: Quantitative comparisons of intensity images from different methods under various scanning grids on the synthetic test set. The output spatial resolution is 128×128. The best is in bold.

| Points | Metrics | FBP [11] | LCT [12] | FK [14] | RSD [15] | UNet [28] | NeTF [35] | LFE [5] | CSA [17] | Ours |
|--------|---------|----------|----------|---------|----------|-----------|-----------|---------|----------|------|
| 32×32 | PSNR(dB) | 21.12 | 22.07 | 24.59 | 25.44 | 26.71 | 22.85 | 27.94 | 23.22 | **28.64** |
|  | ACC(%) | 0.52 | 56.54 | 15.48 | 69.09 | 73.71 | 68.25 | 73.05 | 30.43 | **74.35** |
|  | SSIM | 0.2512 | 0.6314 | 0.4152 | 0.7847 | 0.8808 | 0.8691 | 0.8867 | 0.8223 | **0.8975** |
| 16×16 | PSNR(dB) | 15.69 | 20.27 | 18.02 | 23.53 | 26.64 | 19.14 | 27.04 | 22.62 | **28.00** |
|  | ACC(%) | 0.05 | 20.16 | 5.99 | 60.86 | 73.79 | 62.38 | 72.40 | 25.83 | **74.19** |
|  | SSIM | 0.0851 | 0.4335 | 0.2121 | 0.7102 | 0.8827 | 0.7977 | 0.8620 | 0.8064 | **0.8929** |
| 8×8 | PSNR(dB) | 12.31 | 15.85 | 17.20 | 19.77 | 26.28 | 17.03 | 26.10 | 21.46 | **27.30** |
|  | ACC(%) | 0.031 | 2.64 | 7.21 | 32.54 | 73.56 | 54.87 | 72.17 | 15.56 | **73.59** |
|  | SSIM | 0.0441 | 0.1981 | 0.2147 | 0.5131 | 0.8775 | 0.7639 | 0.8218 | 0.7976 | **0.8789** |

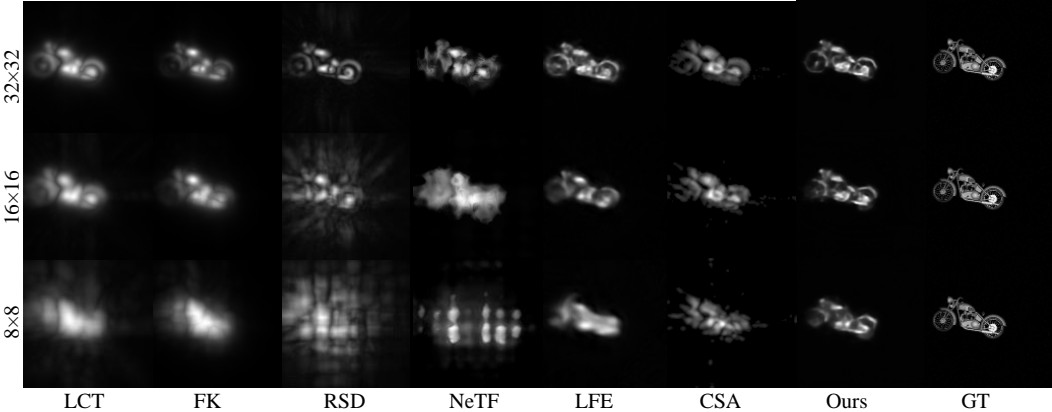

| LCT | FK | RSD | NeTF | LFE | CSA | Ours | GT |

Figure 4: Qualitative comparisons of intensity images from different methods under various scanning grids on the synthetic test set. The final spatial resolution is 128×128.

### 5.3.2 Volume Reconstruction

**Synthetic Results.** The quantitative results on the synthetic test set are reported in Table 1. As the number of scanning points reduces, the traditional methods' metrics decrease significantly, while the deep-learning-based methods except NeTF [35] perform more robustly. Moreover, our method outperforms the deep methods under different scanning grids in terms of all metrics. The qualitative

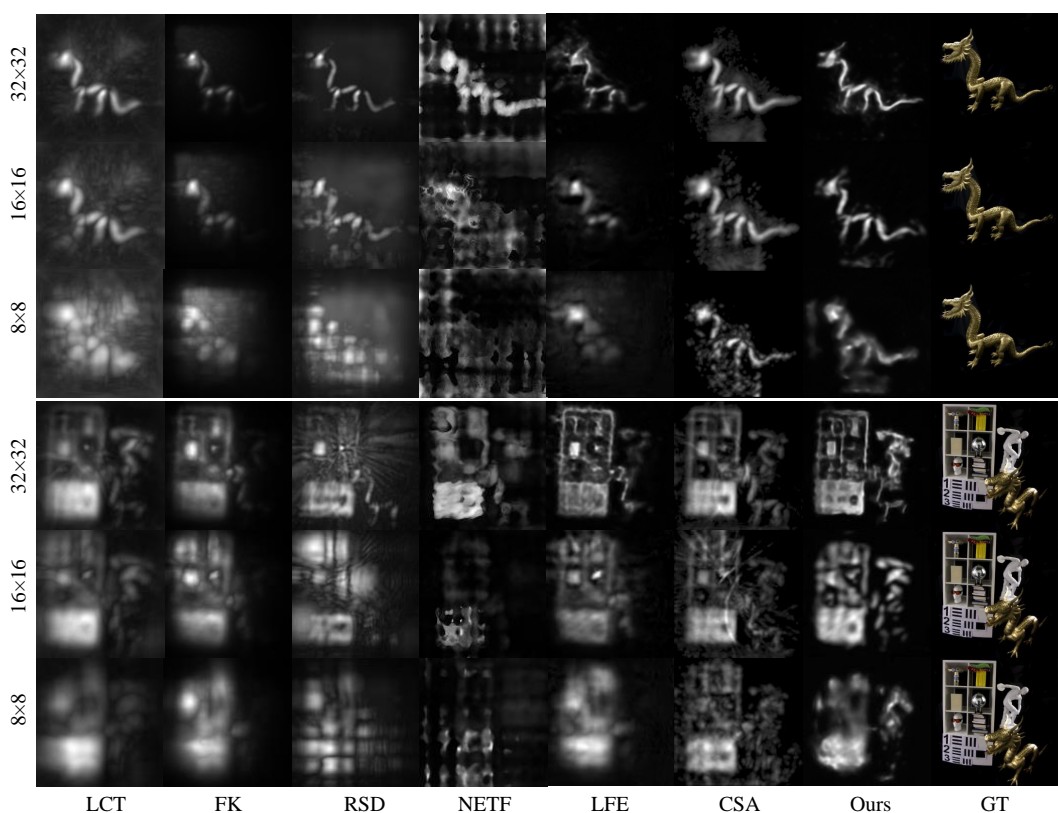

Figure 5: Real-world scenes captured by [14] through three different scanning grids are provided.

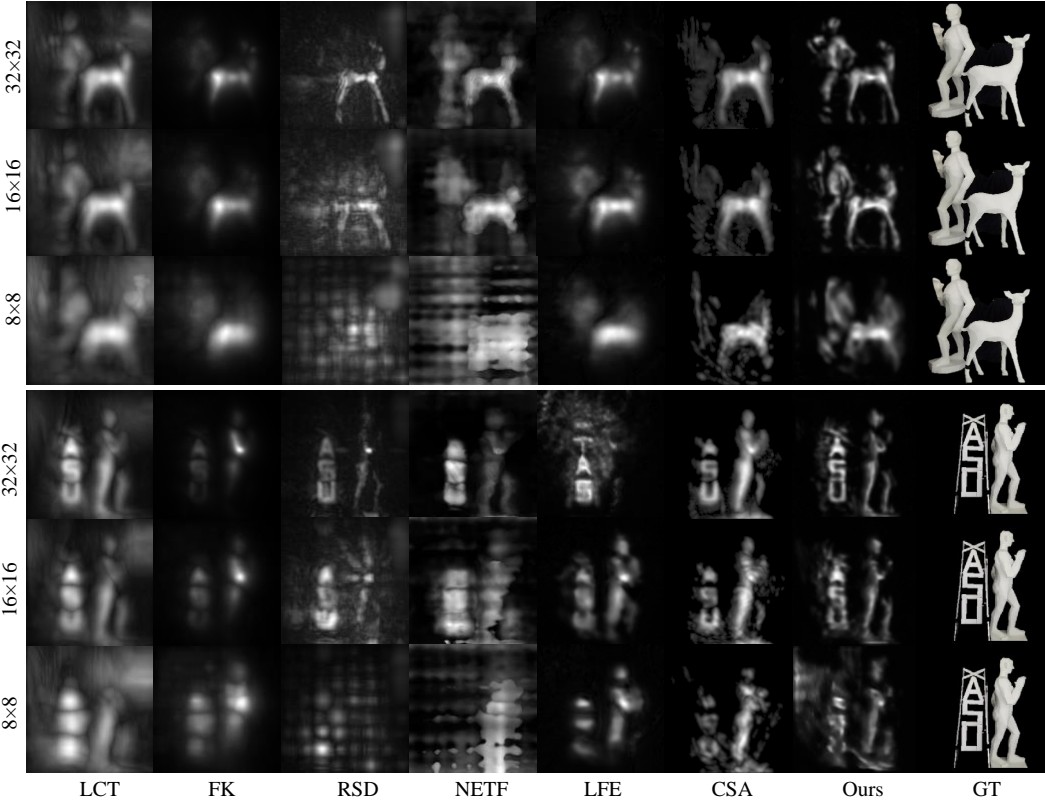

Figure 6: Real-world scenes captured by [29] through three different scanning grids are provided.

Table 2: The inference time and memory of different models. Note that only CSA and our method are specifically designed for NLOS imaging from USM.

| Method | FBP [11] | LCT [12] | FK [14] | RSD [15] | UNet [28] | NeTF [35] | LFE [5] | CSA [17] | Ours |
|---|---|---|---|---|---|---|---|---|---|
| Time (s) | 0.042 | 0.034 | 0.061 | 0.038 | 0.162 | 2h-24h | 0.030 | 20 | 0.420 |
| Memory (M) | 6026 | 6016 | 8056 | 10344 | 5642 | 2G-23G | 4692 | 5306 | 7130 |

results are presented in Fig. 4. The traditional methods fail to reconstruct the motorbike under low scanning points, particularly under the 8×8 scanning grid. Alternatively, LFE [5] and CSA [17] generate the overall structure with textureless information, whereas our method captures more details even with few scanning points. See more qualitative results in the supplement.

**Real-world Results.** In Fig. 5 and Fig. 6, we provide qualitative results in the real-world scenes. We exclude FBP [11] and UNet [28] from the results as they cannot perform well under under-scanning conditions. Specifically, under the 32×32 and 16×16 scanning grids, the traditional methods, LCT [12], FK [14], and RSD [15], support reconstruction with heavy noise. Meanwhile, NeTF [35] fails under the 16×16 scanning grid. LFE [5] provides results with lower noise than traditional methods but fails to capture several details such as the tail of the Dragon and the statue near the bookshelf in Fig. 5. By contrast, CSA [17] and our proposed method produce complete and clear results, with our method generally capturing more details and producing less noise, such as the horns of the dragon and the torso of the statue. Under the 8×8 scanning grid, all baselines degrade significantly. Nonetheless, both CSA and our method achieve satisfactory results. Our method performs better than CSA [17] in most scenes and achieves comparable results to CSA about the bookshelf. It is worth noting that our proposed method offers faster inference time than CSA, which will be discussed in Sec 5.4. More qualitative results of real-world transient measurements, which are captured by different imaging systems and under different total acquisition time, are reported in the supplement.

## 5.4 Inference Time

We conduct experiments to evaluate the inference time and memory requirements of various methods. Specifically, we utilize an NVIDIA GeForce A100 GPU for all testing experiments. Traditional methods including FBP [11], LCT [12], FK [14], and RSD [15] are reimplemented using PyTorch. To accommodate CSA [17], which was originally implemented in Matlab on the CPU, we modify the code to enable computations on the GPU, resulting in a significant acceleration. Table 2 presents the comprehensive results.

The traditional methods and LFE [5] achieve a real-time level of performance. NeTF [35] renders each measurement and requires a huge computational time. However, these methods can not be applied to NLOS reconstruction from USM well. CSA [17], the existing solution for NLOS imaging from USM, suffers from iterative computing times and is not well-suited for practical use. By contrast, our method leverages recent advancements in deep learning, enabling it to perform a single forward pass that takes only 420ms (370ms for transient recovery and 50ms for volume reconstruction). This is nearly 50 times faster than CSA [17]. Moreover, our method does not require iterative computing, which is a significant advantage over CSA [17]. Additionally, our method has a reasonable inference memory requirement, feasible for deployment on portable devices to explore real-world scenarios.

## 5.5 Ablation Study

In this section, we assess the effectiveness of the proposed TRN, VRN, and the regularized loss items: $\mathcal{L}_{ls}$ and $\mathcal{L}_{tv}$, respectively. Additionally, we investigate the effectiveness of the kernel size of MKFE in TRN. All the ablation experiments are conducted under the 16×16 scanning grid condition. We quantitatively evaluate the results on a synthetic test set, while the qualitative results are tested on real-world data.

Table 3 lists the ablation results of the proposed TRN, VRN, and the regularized loss items. Initially, we combine the proposed TRN with LFE [5], and the metrics surpass the original LFE [5], which demonstrates the effectiveness of TRN. Subsequently, we combine the interpolation operator and the proposed VRN. The metrics surpass the original LFE again, which demonstrates the effectiveness of VRN. When we integrate the proposed TRN and VRN, the metrics improve notably. Finally, we

Table 3: Ablation study for TRN, VRN, and regularized loss items. TR and VR denote the transient recovery and volume reconstruction components.

|  | TR | VR | $\mathcal{L}_{ls,tv}$ | PSNR | SSIM | ACC |
|---|---|---|---|---|---|---|
| (-) | Interp | LFE [5] | $\times$ | 27.04 | 0.8620 | 72.40 |
| (a) | TRN | LFE [5] | $\times$ | 27.79 | 0.8692 | 72.75 |
| (b) | Interp | VRN | $\times$ | 27.71 | 0.8646 | 72.85 |
| (c) | TRN | VRN | $\times$ | **28.00** | 0.8899 | 73.44 |
| (d) | TRN | VRN | $\checkmark$ | **28.00** | **0.8929** | **74.19** |

Table 4: Ablation study for the kernel size in multiple kernel feature extraction module of TRN.

| Kernel Size | PSNR | SSIM | ACC |
|---|---|---|---|
| 3 | 27.80 | 0.8744 | 73.98 |
| 5 | 27.89 | 0.8791 | 73.87 |
| 9 | 27.92 | 0.8825 | 73.91 |
| 17 | 27.95 | 0.8696 | 73.85 |
| 3,5,9,17 | **28.00** | **0.8929** | **74.19** |

| (a) | (b) | (c) | (d) | (a) | (b) | (c) | (d) |

Figure 7: Reconstructed intensity images of the models, whose indexes correspond to Table 3.

introduce the regularized loss items: $\mathcal{L}_{ls}$ and $\mathcal{L}_{tv}$. As a result, the SSIM and ACC metrics demonstrate further improvement. Qualitative results corresponding to Table 3 can be seen in Fig. 7.

Table 4 presents the ablation results of various kernel sizes used in MKFE. It can be observed that the PSNR metric tends to increase along with the kernel size, whereas the SSIM and ACC metrics do not mirror that trend. This can be attributed to the fact that the larger kernel could recognize distant regions, while the smaller one could focus on local areas. Notably, the multiple-kernel mechanism results in the best performance across all metrics.

# 6 Conclusions and Limitations

**Conclusions.** In this paper, we study the challenging task of non-line-of-sight imaging from the under-scanning measurement in the deep learning perspective. The elaborate network, which is composed of two components: a transient recovery network and a volume reconstruction network, is proposed to recover the transient measurement and reconstruct the hidden volume successfully. Further, the regularized constraints are exploited to address overfitting and enhance more local details while suppressing smoothing. Notably, our method runs 50 times faster than the existing iterative solution. We believe that our method is a significant step forward in the field of non-line-of-sight imaging and has the potential to unlock new possibilities for future research.

**Limitations.** Several limitations exist in our proposed method, which serve as subjects for future work. Firstly, our forward model is constrained to a linear confocal system, our future work involves generalizing the forward model. Secondly, our proposed method is fully supervised, which may restrain the generalization ability for real-world data. Designing an unsupervised training method for non-line-of-sight imaging is another future work.

# Acknowledgement

This work was supported in part by the National Natural Science Foundation of China under Grants 62131003 and 61901435, and Innovation Program for Quantum Science and Technology under Grant 2021ZD0300300.

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
