# Supplementary Material for
# Deep Non-line-of-sight Imaging
# from Under-scanning Measurements

**Yue Li**  **Yueyi Zhang**[*]  **Juntian Ye**  **Feihu Xu**  **Zhiwei Xiong**
University of Science and Technology of China
{yueli65,jt141884}@mail.ustc.edu.cn
{zhyuey,feihuxu,zwxiong}@ustc.edu.cn

In this supplementary document, we provide more details about the observation model and we explore the trade-off between the number of samplings and noise level for the final reconstruction quality. Furthermore, we present diverse qualitative results from the real-world transient measurements which are captured by different imaging systems. Finally, we provide the quantitative results from the synthetic test data with domain gap to comment on the overfitting risk of the method.

## 1    Observation Model

The active confocal NLOS imaging observation model can be expressed as follows:

$$\tau(\mathbf{s}, t) = \iiint_{\Omega} \rho(\mathbf{o}) \frac{\delta\left(2||\mathbf{s} - \mathbf{o}|| - c \cdot t\right)}{||\mathbf{s} - \mathbf{o}||^4} d\Omega, \tag{1}$$

where $\tau$ denotes the transient measurement, $\Omega$ denotes the spherical waves of the scattered light, $\rho$ represents the volume of the albedo of the hidden object surface, $\delta$ models the light propagation function, $||\mathbf{s} - \mathbf{o}||^4$ is the quartic radiance falloff of light transport between the relay wall and the object, and $c$ is the speed of the light.

As discussed in [1, 2], Eq. 1 can be derived to a straightforward 3D convolution form via variables changing:

$$R_t(\tau) = h \otimes R_z(\rho) \xrightarrow{\text{Discretize}} \boldsymbol{R_t}\boldsymbol{\tau} = \boldsymbol{H}\boldsymbol{R_z}\boldsymbol{\rho} \longrightarrow \boldsymbol{\tau} = \boldsymbol{R_t^{-1}}\boldsymbol{H}\boldsymbol{R_z}\boldsymbol{\rho}, \tag{2}$$

where $R_t(\tau)$ resamples $\tau$ along the time axis, $R_z(\rho)$ resamples $\rho$ along the depth axis, $h$ denotes a shift-invariant convolution kernel, $\otimes$ represents the 3D convolution operator, $\boldsymbol{\tau} \in \mathbb{R}^{h_x w_y n_t}$ and $\boldsymbol{\rho} \in \mathbb{R}^{h_x w_y n_z}$ refer to the vectorized transient measurement and the volume of the albedo of the hidden object. $h_x$ and $w_y$ are the number of the scanning points along the $x$ and $y$ axis, $n_t$ and $n_z$ are the length of the discretized histogram bins along time $t$ axis and depth bins along depth $z$ axis. The determination for $\boldsymbol{\rho} = (\boldsymbol{R_t^{-1}}\boldsymbol{H}\boldsymbol{R_z}\boldsymbol{\rho})^{-1}\boldsymbol{\tau} = \boldsymbol{A^{-1}}\boldsymbol{\tau}$ is an inverse problem, involving 3D deconvolution procedures.

## 2    Results from Different Noise Levels

To show the trade-off between the number of samplings and noise level for the final reconstruction quality, we build a confocal NLOS imaging system and execute experiments across varying noise levels. We capture the real-world transient measurements with different acquisition time per point and scanning grids while maintaining the total acquisition time constant.

In Fig. 1, we present the results obtained with a total acquisition time of 25.6s and 102.4s, respectively. Through these experiments, two crucial insights emerge: 1) Under the same total acquisition time,

---

[*]Corresponding author

37th Conference on Neural Information Processing Systems (NeurIPS 2023).

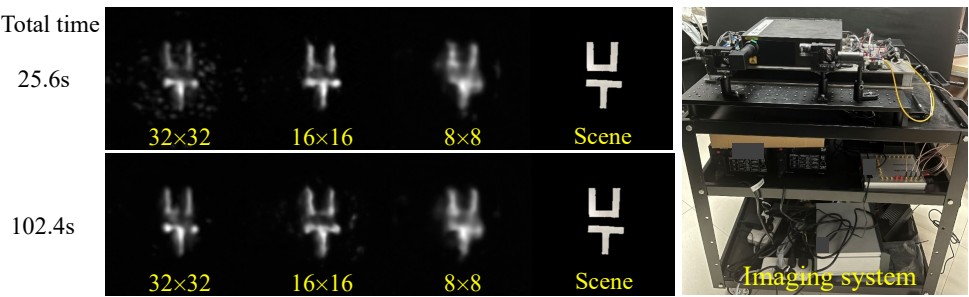

Figure 1: Left: Qualitative results about the noise level experiments. Right: Our NLOS imaging system.

the results obtained from the 16×16 scanning grid are better than those from the 32×32 scanning grid, revealing that longer exposure times for fewer scanning points yield better imaging quality. 2) However, when comparing the results from 32×32 and 8×8 scanning grids, this phenomenon is observed to be less pronounced under extremely low scanning grids.

In light of these insights, we infer that for our specific imaging system, employing a 16×16 scanning grid strikes a balance between the reconstruction quality and the total acquisition time (yet this would vary for different systems). Furthermore, it is noteworthy that our paper introduces a universally applicable technique for NLOS imaging utilizing under-scanning measurements. This method is agnostic to the tradeoff scanning grid of any given imaging system, reaffirming its broad applicability.

**More details about our NLOS imaging system.** Our system operates in a confocal manner, with a 532 nm laser emitting pulses at 70 ps pulse width and 20 MHz repetition frequency. The emitted pulses pass through a two-axis raster-scanning galvo mirror, transmitting onto the visible wall. Subsequently, both direct and indirect diffuse photons are collected by another two-axis galvo mirror and coupled to a multimode fiber, directed towards a free-running single-photon avalanche diode (SPAD) detector with a detection efficiency of approximately 40%. A time-correlated single photon counter records the sync signals from the laser and the photon-detection signals from the SPAD. During data collection, the scanning points and sampling points maintain the same direction but are slightly misaligned to avoid capturing the first bouncing signals during scanning. We employ raster scanning across a 2m × 2m area on the visible wall, and each transient measurement's histogram length is set to 512, with a bin width of 32 ps.

## 3 More Real-world Results

### 3.1 Data captured by [3]

**Firstly**, we provide more real-world results from the transient measurement with a resolution of 128×128×512 and a total acquisition time of 45min. The qualitative results are shown in Fig. 2.

**Secondly**, we provide more real-world results from the transient measurement with a resolution of 128×128×512 and a total acquisition time of 37.5s. A 32×32, 16×16, and 8×8 scanning grid represents the total acquisition time of 2.3s, 0.15s, and 0.009s respectively. The qualitative results are shown in Fig. 3 and Fig. 4. As can be seen, our method still delivers robustness on the transient measurements with a low SNR (Signal to Noise Ratio).

### 3.2 Data captured by our imaging system

To further validate the generalization capability of our model, we conduct tests on additional real-world data captured by our NLOS imaging system. In Fig. 5, we showcase two scenarios for comparison. As evident from the results, our proposed method consistently generates promising outcomes, exhibiting fine details and reduced noise levels. Notably, our method also delivers remarkable robustness even under extremely low scanning grids, highlighting its adaptability to challenging conditions. In contrast to other existing solutions, our method stands out for its superior performance and generalization capabilities across diverse scenes.

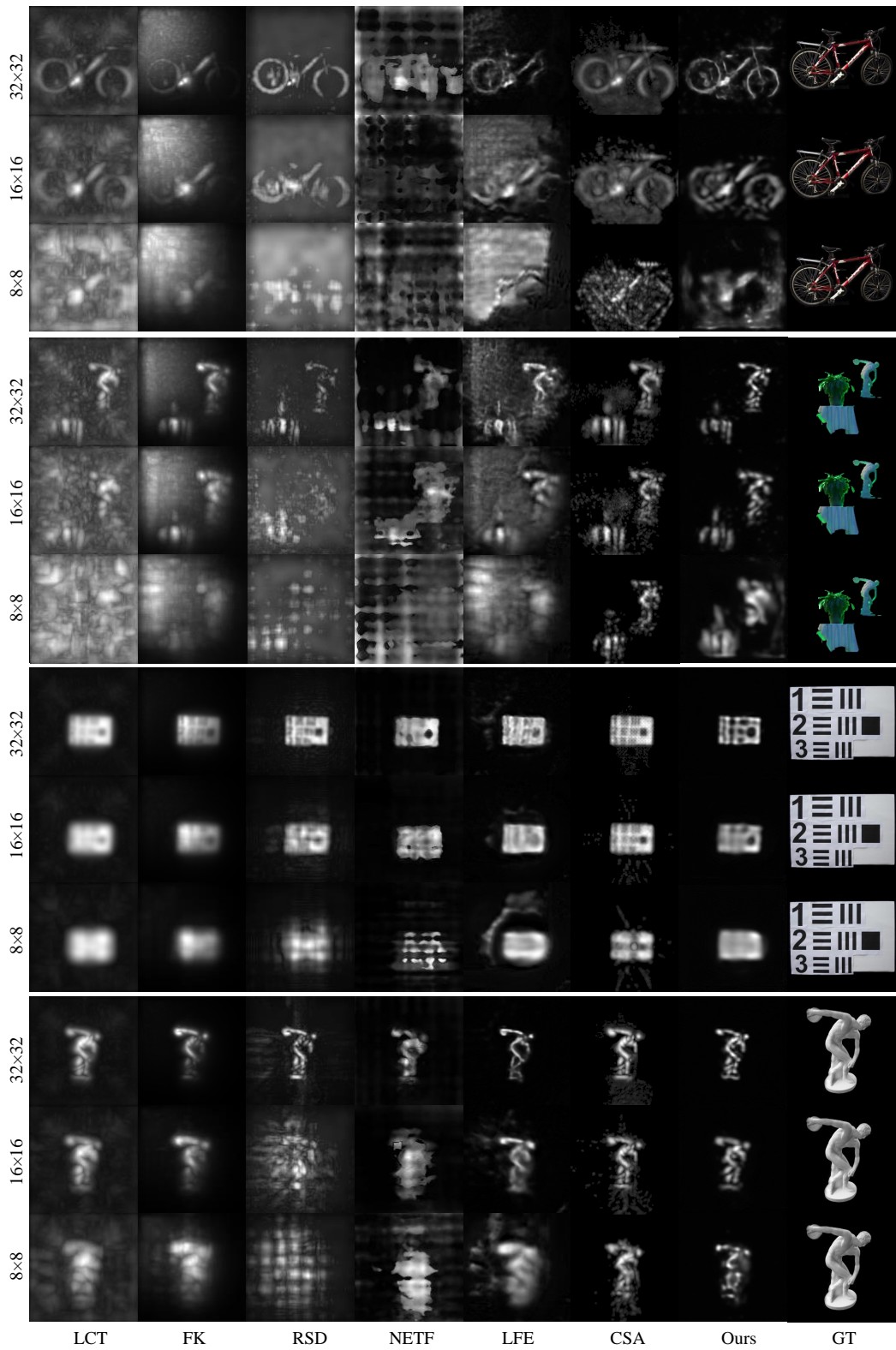

Figure 2: Qualitative comparisons of intensity images from different methods under various scanning grids on the synthetic test set. The final spatial resolution is 128×128. The total acquisition time of the raw transient measurement is **45min**.

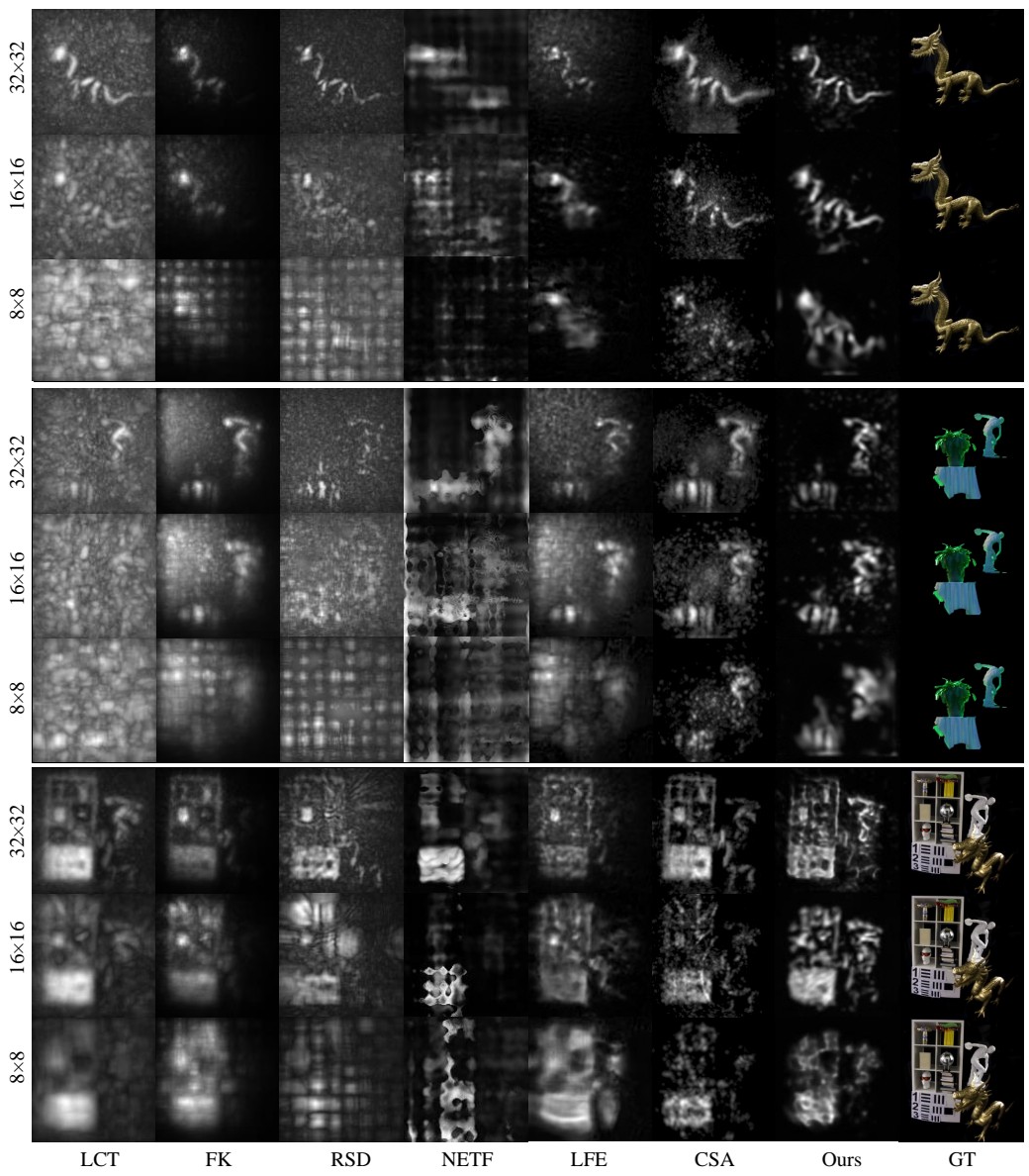

Figure 3: Qualitative comparisons of intensity images from different methods under various scanning grids on the synthetic test set. The final spatial resolution is 128×128. The total acquisition time of the raw measurement is **37.5s**.

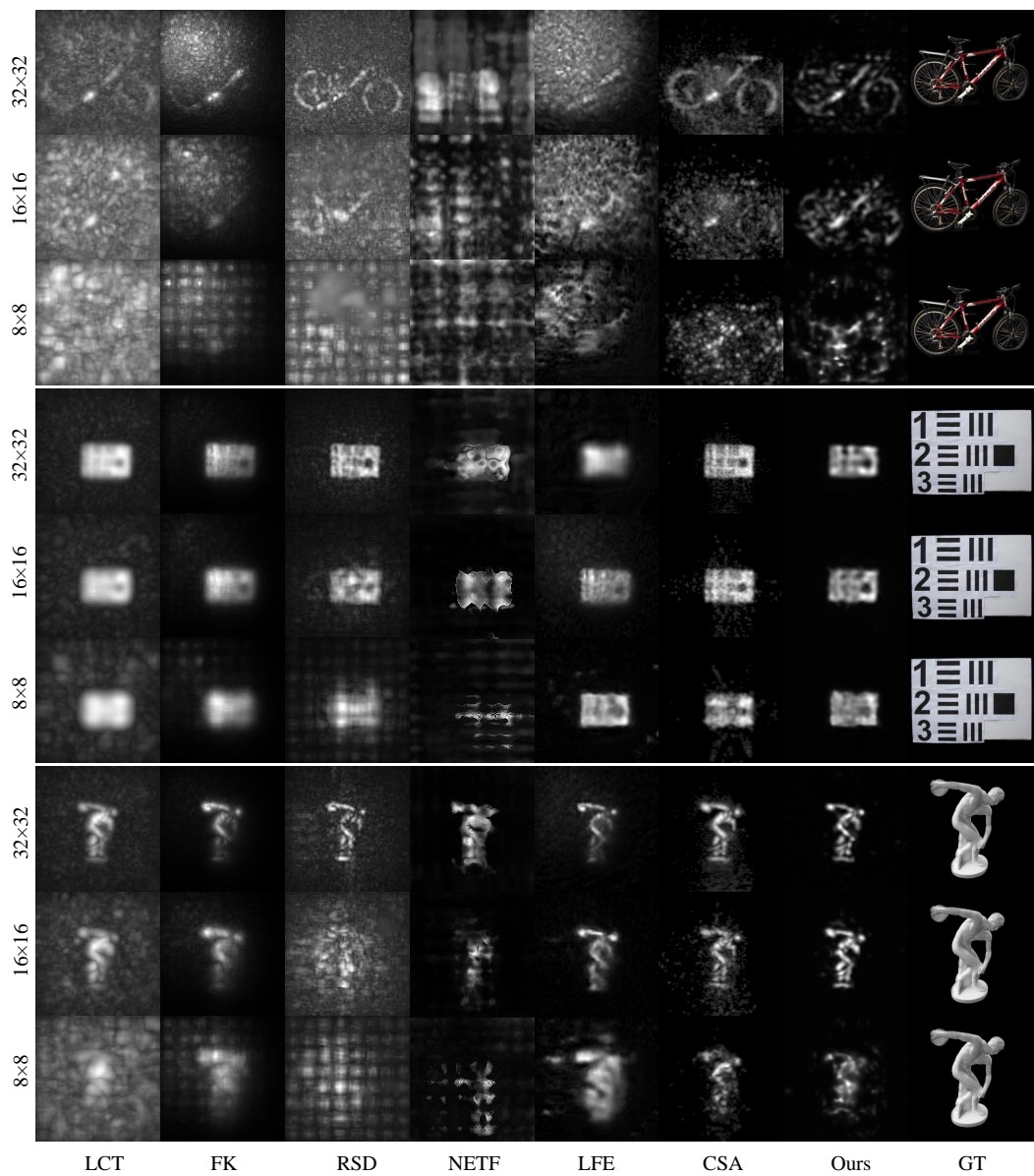

Figure 4: Qualitative comparisons of intensity images from different methods under various scanning grids on the synthetic test set. The final spatial resolution is 128×128. The total acquisition time of the raw transient measurement is **37.5s**.

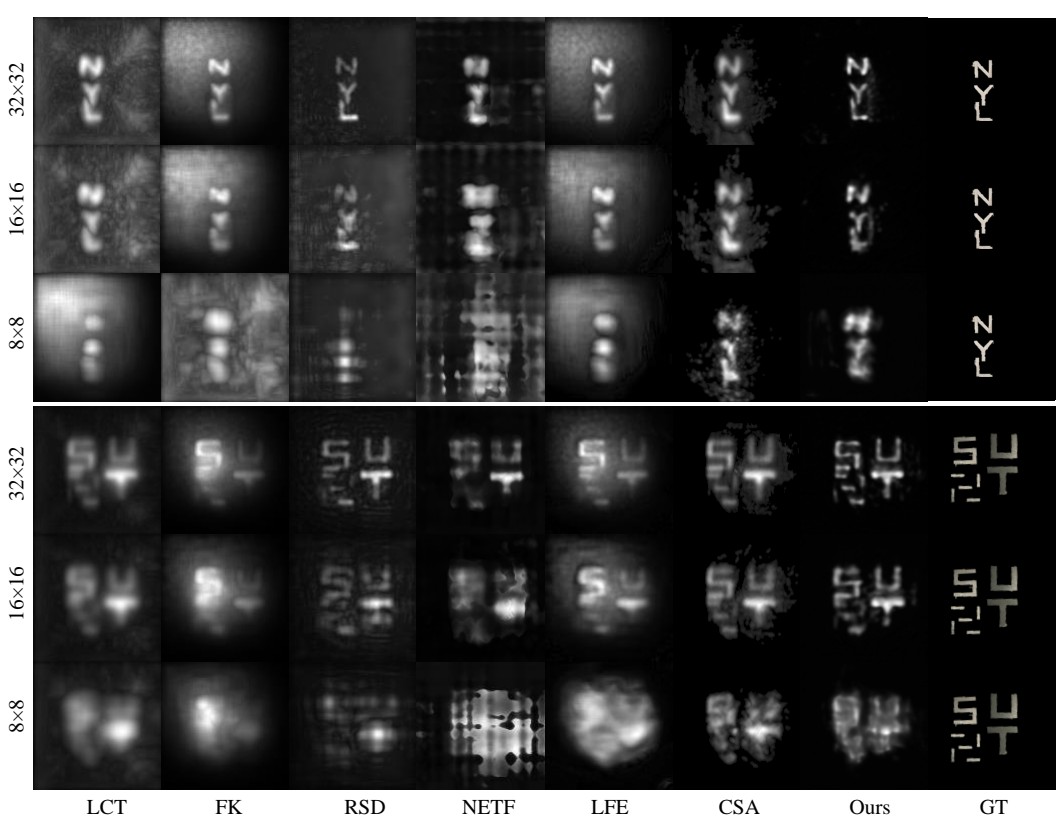

Figure 5: Real-world qualitative results captured by our imaging system, see Fig. 1 right part.

# 4  More Synthetic Results

## 4.1  Data without domain gap

More reconstructed intensity images from the simulated test set are shown in Fig. 6.

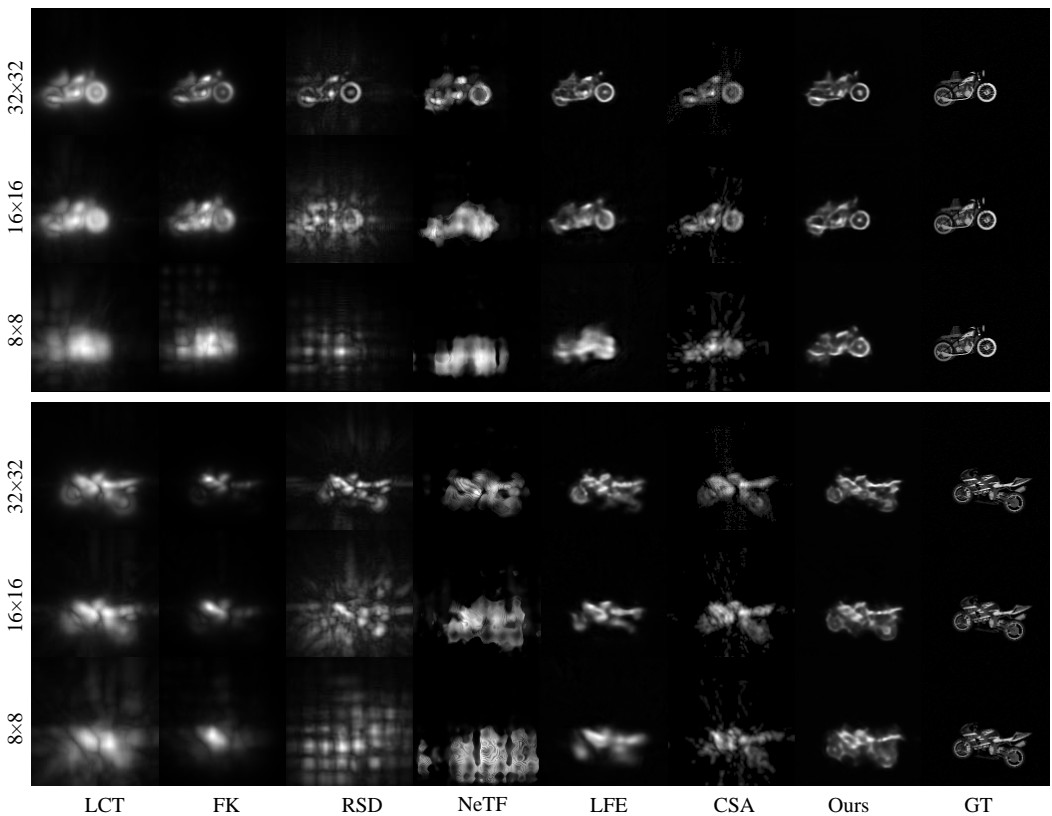

Figure 6: Qualitative comparisons of intensity images from different methods under various scanning grids on the synthetic test set. The final spatial resolution is 128×128.

## 4.2  Data with domain gap

Moreover, we experiment to comment on the overfitting risk of our method. In the paper, the training dataset is rendered from 277 motorbikes. Subsequently, we generate an additional **Unseen** testing dataset by rendering 250 transient measurements from other objects (e.g., baskets, cars, cupboards, etc.). The quantitative results, including NeTF and CSA, are listed in Table 1, and the qualitative results are presented in Fig. 7. As evident from the results, our method outperforms the competition in intensity reconstruction, showcasing the superior generalization capability of our network to unseen objects.

Table 1: Quantitative comparisons of intensity images from different methods under various scanning grids on the **Unseen** synthetic test set. The output spatial resolution is 128×128. The best is in bold.

| Points | Metrics | FBP [4] | LCT [1] | FK [3] | RSD [5] | UNet [6] | NeTF [7] | LFE [8] | CSA [9] | Ours |
|---|---|---|---|---|---|---|---|---|---|---|
| | PSNR(dB) | 18.97 | 20.37 | 26.64 | 23.06 | 26.17 | 24.11 | 28.25 | 26.63 | **30.30** |
| 32×32 | ACC(%) | 4.53 | 53.08 | 72.33 | 23.58 | 74.34 | 64.21 | 70.80 | 43.67 | **76.00** |
| | SSIM | 0.1371 | 0.4815 | 0.8547 | 0.3080 | 0.7510 | 0.8345 | 0.8175 | **0.8617** | 0.8548 |
| | PSNR(dB) | 15.31 | 19.25 | 24.71 | 19.17 | 26.76 | 20.77 | 29.15 | 25.88 | **30.04** |
| 16×16 | ACC(%) | 3.50 | 25.00 | 61.50 | 14.87 | 74.43 | 58.63 | 68.15 | 35.65 | **74.43** |
| | SSIM | 0.0796 | 0.1154 | 0.7320 | 0.2160 | 0.7810 | 0.7661 | 0.7295 | **0.8302** | 0.7980 |
| | PSNR(dB) | 14.80 | 15.45 | 20.49 | 18.63 | 25.11 | 17.87 | 29.19 | 23.65 | **30.01** |
| 8×8 | ACC(%) | 3.71 | 9.30 | 40.18 | 15.00 | 71.68 | 47.44 | 72.76 | 24.13 | **74.00** |
| | SSIM | 0.0726 | 0.1627 | 0.5387 | 0.2170 | 0.7676 | 0.6589 | 0.7440 | 0.7440 | **0.7900** |

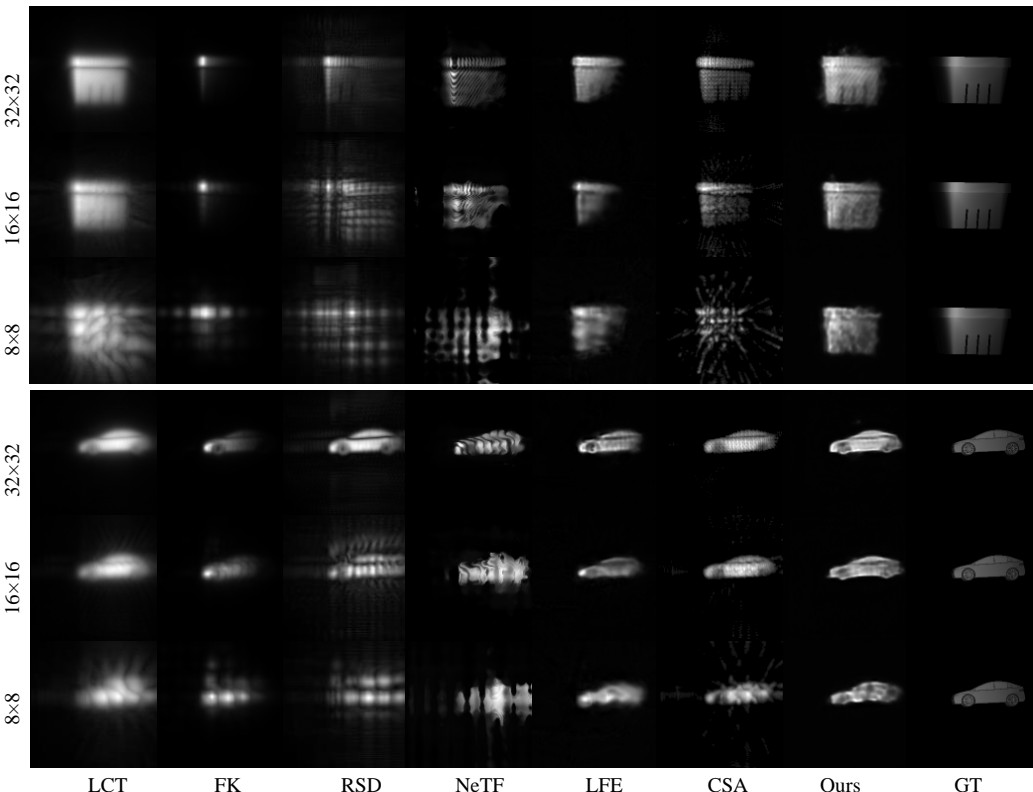

Figure 7: More qualitative results on the **Unseen** synthetic data.