# OpenReview forum: "Deep Non-line-of-sight Imaging from Under-scanning Measurements"
_NeurIPS.cc/2023/Conference — NeurIPS 2023 poster_

### Official Review · Reviewer_VzCp · 2023-07-03

**Soundness:** 3 good
**Presentation:** 3 good
**Contribution:** 2 fair
**Rating:** 5
**Confidence:** 4

**Summary:**

This paper proposes a new deep learning based method for NLOS imaging reconstruction. Their method is an end-to-end deep network that directly acts on under scanning transient measurements to recover the NLOS image. The use of a single pass network enables fast, realtime inference. The key contribution is the extension to under scanning measurement regime, i.e. transient inputs are only available over a sparse grid of scanning points. This is enabled using the “transient recovery network” that performs padding+convolutions using kernels of different sizes to upsample the transient inputs and extract features, and further processing to estimate the sufficient scanning measurements.

These upsampled transients are transformed from xyt space to xyz space in the “volume reconstruction network” (by inverting the LCT, which is essentially a 3D deconvolution operation), and then projected back to 2D image space to recover the final image. The architecture for this component closely resembles the architecture in LFE [5] (which is also an end-to-end NLOS image reconstruction method, but which is trained to work with sufficient scanning transient measurements).

The authors also provide extensive experiment results for different baseline methods (both traditional reconstruction methods which don't rely on any learning across multiple image/transient pairs, and deep learning based methods: LFE and U-net). They demonstrate the superior quality as well as reduced inference time of their method. They evaluate the TRN separately against the baseline of simply interpolating the transient histogram from neighboring points.

**Strengths:**

1. The paper takes an important step towards realizing the goal of real time NLOS imaging. This is enabled by the use of a fast, end-to-end reconstruction algorithm combined with faster measurement (through underscanning).
2. Ablation studies are done well (e.g. contribution of regularization loss terms, contribution of TRN and VRN is quantified individually)
3. Comparison to baseline methods is done in a fair manner, e.g. for the CSA method they reimplement the code to enable acceleration on GPU.
4. Good writing and clear explanation of method and results

**Weaknesses:**

The ML contribution of this paper is limited. While the paper makes good incremental progress on an important real world imaging problem, there weren't any novel insights into the nature of the problem (e.g. the compressive sensing aspect, is there any minimum number of scans needed etc). The proposed architecture is also not very novel either.

**Questions:**

1. Can you comment on any risk of overfitting to the training domain with your method (and deep learning methods in general)? It’s important to quantify that risk since the traditional methods () don’t have any such risk.
2. You mention in lines 226-227 that "Due to their time-intensive nature and no need for training on simulated datasets, we exclude the results of NeTF [33] and CSA [17] from the analysis". I didn't understand the second reason. These methods don't require any training, so it should be easy to include them in the analysis, right? Also if you were able to include these methods in the real-world results, then the first reason (time-intensive nature) also shouldn't hold.

**Limitations:**

Yes

---

> ### Author Rebuttal · Authors · 2023-08-08
>
> We thank the reviewer for giving us valuable comments. The main concerns are addressed below.
>
> **Q1-1:** *The ML contribution of this paper is limited. ... So I'm not sure of the suitability of the paper for a venue like Neurips.*\
> **Reply:** First of all, we would like to kindly remind that our submission is to the track "ML for physical sciences". As recognized by the reviewer, this work "takes an important step towards realizing the goal of real-time NLOS imaging", we therefore believe it is suitable for this track in NeurIPS. For the similar reason, the main novelty of this work does not lie in the architecture design, but adapting existing deep learning techniques to this challenging task with an end-to-end framework, which is nontrivial from the perspective of "AI for Science". Meanwhile, we believe **the strengths of this work are distinct: the comprehensive experiments, thorough ablation, and clearly new state-of-the-art performance have been well recognized by ALL reviewers**. Being confident in our results, we are ready to release the source code to the public. We hope this cross-discipline work, together with its repository, will contribute to the diversity of this conference, which allows other researchers in this scientific field to build on top of it and improve the field further with an enhanced baseline.
>
> **Q1-2:** *Novel insights into the nature of the problem (e.g. the compressive sensing aspect, is there any minimum number of scans needed etc).*\
> **Reply:** As suggested by Reviewer r4bJ, we now add a set of experiments to explore the tradeoff between the number of samplings and noisy level for the final reconstruction quality, which we believe provides novel insights into the nature of the problem. In light of these insights, we infer that for our specific imaging system, employing a 16*16 scanning grid strikes a balance between the reconstruction quality and the total acquisition time (yet this would vary for different systems).  Please refer to the global response for details.
>
> **Q2:** *Comment on any risk of overfitting.*\
> **Reply:**
> As suggested, we have conducted two experiments to comment on the overfitting risk of our method:
>
> 1) Test on the **Unseen** synthetic dataset\
> In the manuscript, the training dataset is rendered from 277 motorbikes. Subsequently, we generate an additional **Unseen** testing dataset by rendering 250 transient measurements from other objects (e.g., baskets, cars, cupboards, etc.). The quantitative results, including NeTF and CSA, are listed in Table 1, and the qualitative results are presented in Fig. 3 of the global rebuttal PDF. As evident from the results, our method outperforms the competition in intensity reconstruction, showcasing the superior generalization capability of our network to unseen objects.
>
> 2) Test on more real-world data from **different** imaging systems\
> To further validate the generalization of our model, we extend the evaluation to include more real-world data captured from different NLOS imaging systems. Fig. 2 of the global rebuttal PDF showcases two scenarios: the top scene captured by [1], and the bottom scene captured by our imaging system ( Details in the global rebuttal response). As observed, our method delivers promising results with fine details and reduced noise levels. Moreover, it exhibits impressive robustness even under extremely low scanning grids. The demonstrated superiority of our method over other solutions underlines its remarkable generalization capability across diverse real-world scenes.
>
> [1] Yue Li, Jiayong Peng, Juntian Ye, Yueyi Zhang, Feihu Xu, and Zhiwei Xiong. Nlost: Non-line-of-sight imaging with transformer. In Conference on Computer Vision and Pattern Recognition, pages 13313–13322, 2023.
>
> **Q3:** *You mention in lines 226-227 that "Due to their time-intensive nature and no need for training on simulated datasets, we exclude the results of NeTF [33] and CSA [17] from the analysis". I didn't understand the second reason.*\
> **Reply:**
> We are sorry that the explanation provided in the manuscript may be unclear.
>
> NeTF is a test-time training method that renders the transient field on each single measurement, and can be directly applied to the individual real-world data without training on a large amount of synthetic data. For the same reason, it is computationally extensive to generate NeTF results on the large amount of synthetic data, since each scene requires a long rendering time. CSA is a traditional method that does not require training on synthetic data. It is also quite slow, so considerable time is needed to generate CSA results on the large amount of synthetic data.
>
> Since real-world results are the most important, we did not list the synthetic results of NeTF and CSA in the original manuscript due to limited time. However, as expected, we now take abundant computing resources to supplement all synthetic results for NeTF and CSA. For NeTF, we train each under-scanning transient measurement under different scanning grids for testing. For CSA, we also evaluate all the test data. The quantitative results for different scanning grids are as follows:
> | PSNR/ACC/SSIM | NeTF               | CSA                | Ours                   |
> | ------------- | ------------------ | ------------------ | ---------------------- |
> | 32*32         | 22.85/68.25/0.8691 | 23.22/30.43/0.8223 | **28.64/74.35/0.8975** |
> | 16*16         | 19.14/62.38/0.7977 | 22.62/25.83/0.8064 | **28.00/74.19/0.8929** |
> | 8*8           | 17.03/54.87/0.7639 | 21.46/15.56/0.7976 | **27.30/73.59/0.8789** |
>
>
> As evident from the results, our proposed method consistently outperforms NeTF and CSA across all scanning grid sizes, achieving higher PSNR, ACC, and SSIM metrics.
>
> We truly appreciate your valuable suggestion and we will revise the explanation. We will include these quantitative results in the manuscript to offer a comprehensive and transparent assessment of our method's capabilities.

---

> ### Comment · Reviewer_VzCp · 2023-08-13
>
> Thanks to the authors for their detailed rebuttal. I really appreciate the additional experiments they conducted (for testing generalization, and the inclusion of the missing comparisons to NeTF/CSA).
>
> Based on the response, I've changed my rating to 5 (Borderline accept). I've also edited my review to remove the statement about non-suitability for Neurips.

---

### Official Review · Reviewer_9P8E · 2023-07-06

**Soundness:** 3 good
**Presentation:** 3 good
**Contribution:** 2 fair
**Rating:** 5
**Confidence:** 4

**Summary:**

1, This paper proposed a new approach to NLOS reconstruction from USM based on deep learning, and achieved high-quality and rapid inference.
2, Two main reconstruction modules, dubbed transient recovery network (TRN) and volume reconstruction network (VRN), are designed to produce the hidden volume and the intensity image at a high-spatial resolution.

**Strengths:**

1, This paper proposed a deep learning-based network for NLOS reconstruction from USM, where the key components are the transient recovery network and the volume reconstruction network. The method is reasonable and is an inspirable attempt to apply such an end-to-end framework to deal with this challenging task.
2, The performance of this method outperforms existing methods both quantitatively and qualitatively, as well as the inference time.

**Weaknesses:**

1, The main modules in this work are heavily based on existing work. For example, the multiple kernel feature extraction block has already been widely applied in many downstream vision tasks, like “Multiple kernel learning for hyperspectral image classification: A review” (Yanfeng Guo, et al. TGRS 2017). Besides, the volume reconstruction network is very similar to LFE (“Learned feature embeddings for non-line-of-sight imaging and recognition”, Wenzheng Chen, et al. ACM TOG 2020). These points indicate that the contribution of this work is not convincing enough, although the performance is satisfying.
2, The details of the volume refinement module are unclear.
3, The loss function in this work is a combination of four loss terms, which may cause generalizability problem to the model, could you please conduct more experiments on more datasets to demonstrate the performance?
4, In Table 2, it shows that LFE costs much less memory and achieves much faster inference time than this method, which indicates the performance of this method is not that outstanding as authors claimed.

**Questions:**

Please refer to Weakness.

**Limitations:**

Please refer to Weakness.

---

> ### Author Rebuttal · Authors · 2023-08-08
>
> We thank the reviewer for giving us valuable comments. The main concerns are addressed below.
>
> **Q1:** *The main modules in this work are heavily based on existing work. These points indicate that the contribution of this work is not convincing enough, although the performance is satisfying.*\
> **Reply:** As we know, one advantage of the AI community is gradually building on top of existing works, which eventually leads to both solid and efficient solutions for many practical applications. When we look back, there are a number of works in this community with seemingly incremental components but have made a great impact on future research. From the perspective of "AI for Science" (the track we submit), the contribution of being ''an inspirable attempt to apply an end-to-end framework to deal with a challenging task'', as recognized by the reviewer, is nontrivial. Meanwhile, we believe **the strengths of this work are distinct: the comprehensive experiments, thorough ablation, and clearly new state-of-the-art performance have been well recognized by ALL reviewers**. Being confident in our results, we are ready to release the source code to the public. We would appreciate the opportunity to present this work as the new state-of-the-art together with its repository at this conference, which allows other researchers in this scientific field to build on top of it and improve the field further with an enhanced baseline.
>
> **Q2:** *The details of the volume refinement module are unclear.*\
> **Reply:** The volume refinement module is constructed with a composition of three 3D convolutions and three interlaced residual layers. Each of these residual layers is composed of two 3D convolutions, followed by a ReLU operation, and ultimately a residual connection. This module has been purposefully designed to proficiently accentuate essential features while amplifying the 3D volume. The objective is to enhance fidelity, thereby avoiding a direct projection of the 3D volume onto 2D planes which could potentially result in the loss of crucial inherent features.
>
> Thanks for your reminder, and we will update this component in detail.
>
> **Q3:** *Could you please conduct more experiments on more datasets to demonstrate the performance?*\
> **Reply:** To further validate the generalization capability of our model, we conduct tests on additional real-world data captured from **different** NLOS imaging systems. In Fig. 2 of the global rebuttal PDF, we showcase two scenarios for comparison. The top scene is captured by [1], while the bottom scene is captured using **our imaging system** (Details in the global response). As evident from the results, our proposed method consistently generates promising outcomes, exhibiting fine details and reduced noise levels. Notably, our method also delivers remarkable robustness even under extremely low scanning grids, highlighting its adaptability to challenging conditions. In contrast to other existing solutions, our method stands out for its superior performance and generalization capabilities across diverse scenes.
>
> [1] Yue Li, Jiayong Peng, Juntian Ye, Yueyi Zhang, Feihu Xu, and Zhiwei Xiong. Nlost: Non-line-of-sight imaging with transformer. In Conference on Computer Vision and Pattern Recognition, pages 13313–13322, 2023.
>
> **Q4:** *In Table 2, it shows that LFE costs much less memory and achieves much faster inference time than this method, which indicates the performance of this method is not that outstanding as authors claimed.*\
> **Reply:** We respectfully disagree with the provided analysis. We would like to address the following three insights:
>
> 1) Among all the solutions considered, only CSA and our method are specifically designed for NLOS imaging from under-scanning measurements (USM). It is important to note that LFE is trained to work with sufficient scanning transient measurements, making it less suitable for USM-based NLOS imaging scenarios.
>
> 2) As mentioned in Line 261 of the manuscript, our method's total inference time is 420ms (370ms for transient recovery and 50ms for volume reconstruction). On the other hand, the inference time of LFE, which solely involves volume reconstruction, is reported to be 30ms. There is a difference of only 20ms between the two methods for volume reconstruction.
>
> 3) In Table 3(b) of the manuscript, the combined approach of our transient recovery network and LFE exhibits improved metrics for PSNR/ACC/SSIM, increasing from 27.04/72.40/0.8620 to 27.79/72.75/0.8692. This improvement indeed demonstrates that there is room for enhancing LFE's performance further.
>
> In summary, while LFE shows less memory and faster inference time, our method stands out as a specialized solution designed explicitly for NLOS imaging from under-scanning measurements.  Furthermore, the combination of our transient recovery network with LFE showcases the potential for further improving LFE's results.

---

> > ### Comment · Reviewer_9P8E · 2023-08-20
> >
> > Thanks for the response which partly addressed our concerns. Please update Q2 in the revised paper.

---

### Official Review · Reviewer_iKMu · 2023-07-07

**Soundness:** 2 fair
**Presentation:** 2 fair
**Contribution:** 3 good
**Rating:** 5
**Confidence:** 3

**Summary:**

The paper proposes a deep learning based approach to non-line-of-sight (NLOS) imaging from under-scanning measurements (USM). This is achieved by first using a transient recovery network (TRN) that increases the spatial resolution of the input, and a volume reconstruction network (VRN) that uses linear physics prior for volume reconstruction and intensity image generation. The method uses local similarity and total variation loss to further improve the reconstruction quality.

**Strengths:**

The proposed approach results in shorter scan time and shows better reconstruction performance at multiple grid resolution compared to all the other methods.

**Weaknesses:**

-The local similarity and TV losses represent surface smoothness and cause problems with holes. What is the motivation behind using a Guassian weighting instead of bilateral weight which preserves discontinuity.

-There are details missing on the real-world data train/test split.


[Minor] Line 128: Fig. 2f should be Fig. 2e


**Questions:**

How is the supervised training done on the real-world dataset [14]?
How would the method scale to a larger real-world dataset?


**Limitations:**

Main limitation as pointed out in the paper is the supervised training approach. For diverse set of scenes the method will need to generalize for arbitrary shapes. The regularization losses can help with smooth surfaces but will not perform well on objects with holes unless the hyperparameters are tuned carefully.

---

> ### Author Rebuttal · Authors · 2023-08-08
>
> We are highly encouraged by the positive recommendation and comments from the reviewer. The main concerns are addressed below.
>
> **Q1:** *What is the motivation behind using a Gaussian weighting instead of bilateral weight which preserves discontinuity?*\
> **Reply:** Indeed, bilateral weights are commonly used in computer vision tasks to achieve denoising while preserving image discontinuity when the processed results contain sufficient information. However, in the context of NLOS reconstruction from under-scanning measurements, the primary challenge lies in reconstructing more hidden volume from the scarce input and the secondary goal is to suppress noise. To address this challenge, we take into account the inherent similarity between neighboring pixels in NLOS scenes. We first leverage the Gaussian weighted NLM term to learn and reconstruct more intricate details in the hidden volume. This step aims to capture and preserve the fine features that might be otherwise lost during the reconstruction process. Subsequently, we apply the TV term to denoise the reconstructed results.
>
> The effectiveness of this approach has also been demonstrated in other areas, including compressed sensing tasks [1,2] and NLOS imaging tasks [3,4]. The success of this methodology in these diverse domains further reinforces its suitability.
>
> [1] Weisheng Dong, Lei Zhang, Guangming Shi, and Xiaolin Wu. Image deblurring and super-resolution by adaptive sparse domain selection and adaptive regularization. IEEE Transactions on image processing, 20(7):1838–1857, 2011.\
> [2] Kan Chang and Baoxin Li. Joint modeling and reconstruction of a compressively-sensed set of correlated images. Journal of Visual Communication and Image Representation, 33:286–300, 2015.\
> [3] Xintong Liu, Jianyu Wang, Zhupeng Li, Zuoqiang Shi, Xing Fu, and Lingyun Qiu. Non-line-of-sight reconstruction with signal–object collaborative regularization. Light: Science & Applications, 10(1):198,2021.\
> [4] Jianjiang Liu, Yijun Zhou, Xin Huang, Zheng-Ping Li, and Feihu Xu. Photon-efficient non-line-of-sight364
> imaging. IEEE Transactions on Computational Imaging, 8:639–650, 2022.
>
> **Q2:** *There are details missing on the real-world data train/test split. How is the supervised training done on the real-world dataset [14]? How would the method scale to a larger real-world dataset?*\
> **Reply:** This could be a misunderstanding. The model is NOT trained on real-world data. Instead, our model is first trained in a supervised manner on the synthetic dataset and then directly test on the real-world data. In principle, there would be no difficulty to scale to a larger real-world dataset ( See experiments in Q3 Reply).
>
> **Q3:** *Main limitation as pointed out in the paper is the supervised training approach. For diverse set of scenes the method will need to generalize for arbitrary shapes.*\
> **Reply:** To further validate the generalization capability of our model, we conduct tests on additional real-world data captured from **different** NLOS imaging systems. In Fig. 2 of the global rebuttal PDF, we showcase two scenarios for comparison. The top scene is captured by [1], while the bottom scene is captured using **our imaging system** ( Details in the global rebuttal response). As evident from the results, our proposed method consistently generates promising outcomes, exhibiting fine details and reduced noise levels. Notably, our method also delivers remarkable robustness even under extremely low scanning grids, highlighting its adaptability to challenging conditions. In contrast to other existing solutions, our method stands out for its superior performance and generalization capabilities across diverse scenes.
>
> [1] Yue Li, Jiayong Peng, Juntian Ye, Yueyi Zhang, Feihu Xu, and Zhiwei Xiong. Nlost: Non-line-of-sight imaging with transformer. In Conference on Computer Vision and Pattern Recognition, pages 13313–13322, 2023.
>
> **Q4:** *Line 178: Fig. 2f should be Fig. 2e*\
> **Reply:**  Thanks for pointing out this typo, we will update it.

---

### Official Review · Reviewer_r4bJ · 2023-07-27

**Soundness:** 3 good
**Presentation:** 3 good
**Contribution:** 2 fair
**Rating:** 5
**Confidence:** 2

**Summary:**

This paper introduces a deep-learning-based method for non-line-of-sight (NLOS) imaging from under-scanning measurements (USM). It comprises two main components: the transient recovery network (TRN) and the volume reconstruction network (VRN). The TRN extracts features and produces high-resolution measurements, while the VRN reconstructs the hidden volume. The method achieves superior performance on synthetic and real-world data, displaying impressive robustness even at an 8x8 scanning grid and providing high-speed inference, 50 times faster than existing solutions.

**Strengths:**

(1) The paper deals with under-scanning measurements and develops an end-to-end network for high-quality and fast inference.
(2) The experiments validate the claim. The experimental results show improvements in both qualitative (Fig. 4, and Fig. 5) and quantitative  (Tab. 1) aspects, and fast inference (Tab. 2). The supplementary material contains further qualitative results, providing additional evidence of the effectiveness of the proposed method.
(3) Ablation is quite comprehensive.

**Weaknesses:**

(1) The paper does not mention how noise is modeled.
(2) It is better to show the tradeoff between the number of samplings and noisy level for the final reconstruction quality.
In the paper, we see that 32x32 sampling is always better than 8x8.
But given the same total acquisition time, 32x32 might be worse than 8x8 under some noise level.

**Questions:**

Please see the weakness part

---

> ### Author Rebuttal · Authors · 2023-08-08
>
> We are highly encouraged by the positive recommendation and comments from the reviewer. The main concerns are addressed below.
>
> **Q1:** *The paper does not mention how noise is modeled.*\
> **Reply:**  For the noise modeling of the NLOS simulation, the formulation is as follows:\
> $$ \hat{\tau}(\mathbf{s},t) \sim \text { Poisson }(\tau(\mathbf{s},t) + B), $$\
> where $\tau$ denotes the raw transient measurement, $\hat{\tau}$ means the resulting transient measurement, and $B$ represents the background photons and dark counts. For the data synthesis, we exclude cross-talk and afterpulsing effects, as [1]. The implementation of our noise model adheres closely to the methodology outlined in [2].\
> We sincerely appreciate your kind reminder and we will explain the noise model in the manuscript.
>
> [1] Wenzheng Chen, Fangyin Wei, Kiriakos N Kutulakos, Szymon Rusinkiewicz, and Felix Heide. Learned feature embeddings for non-line-of-sight imaging and recognition. ACM Transactions on Graphics, 39(6):1–18, 2020.\
> [2] Quercus Hernandez, Diego Gutierrez, and Adrian Jarabo. A computational model of a single-photon
> avalanche diode sensor for transient imaging. arXiv preprint arXiv:1703.02635, 2017.
>
> **Q2:** *Tradeoff between the number of samplings and noisy level for the final reconstruction quality.*\
> **Reply:** As suggested, we build a confocal NLOS imaging system (Details in the global rebuttal response) and execute experiments across varying noise levels. We have captured the real-world transient measurements with different acquisition time per point and scanning grids while maintaining the total acquisition time constant.
>
> In Fig. 1 (left part of the global rebuttal  PDF), we present the results obtained with a total acquisition time of 25.6s and 102.4s, respectively. Through these experiments, two crucial insights have emerged: 1) Under the same total acquisition time, the results obtained from the 16 * 16 scanning grid are better than those from the 32 * 32 scanning grid, revealing that longer exposure times for fewer scanning points yield better imaging quality. 2) However, when comparing the results from 32 * 32 and 8 * 8 scanning grids, this phenomenon is observed to be less pronounced under extremely low scanning grids.
>
> In light of these insights, we infer that for our specific imaging system, employing a 16*16 scanning grid strikes a balance between the reconstruction quality and the total acquisition time (yet this would vary for different systems). Furthermore, it is noteworthy that our manuscript introduces a universally applicable technique for NLOS imaging utilizing under-scanning measurements. This method is agnostic to the tradeoff scanning grid of any given imaging system, reaffirming its broad applicability.

---

> > ### Comment · Reviewer_r4bJ · 2023-08-17
> > **Not satisfied with reply of Q2**
> >
> > For different total acquisition time, the best sampling grid will be different.
> > If the sensor changes (noise level changes), this best sampling grid may also change.
> > However, the authors' reply is superficial.

---

> > > ### Author Response · Authors · 2023-08-17
> > > **Further clarification on the suggested tradeoff**
> > >
> > > Thanks for the comment. We appreciate the opportunity for further clarification.
> > >
> > > 1)	We do agree with the reviewer that there exists a tradeoff between the sampling grid and the noise level, which provides additional insights into the task itself and helps strengthen our work. During the limited rebuttal time, we tried our best to conduct a set of experiments that demonstrate this kind of tradeoff to a certain extent. Although the experiments may not be comprehensive enough, we think the point as suggested by the reviewer has been validated in principle.
> > >
> > > 2)	Despite of the existence of the tradeoff, and its dependency on multiples factors (e.g., total acquisition time, noise level, system configuration, etc.), we would like to emphasize that the proposed method provides a general way to actually enable the best sampling grid (as long as it is an undersampled one) with promising reconstruction performance. This contribution is independent of the above tradeoff, and is what we mainly claim in the original paper.
> > >
> > > 3)	We promise to make a more thorough discussion in the updated version on the suggested tradeoff, not only supplement the results in the rebuttal (and more hopefully), but also provide more serious analysis on this tradeoff by considering its multiple dependencies.
> > >
> > > Thanks again for bringing this insightful suggestion.

---

### Author Rebuttal · Authors · 2023-08-09

# This is the global rebuttal for all the reviewers including a PDF containing only figures. Please find individual responses to each reviewer below.

The content is:

**Fig 1**: Qualitative results about the noise level experiments and our imaging system.

As suggested by Reviewer r4bJ, we build a confocal NLOS imaging system and execute experiments across varying noise levels. We have captured the real-world transient measurements with different acquisition time per point and scanning grids while maintaining the total acquisition time constant.

In Fig. 1 (left part of the global rebuttal PDF), we present the results obtained with a total acquisition time of 25.6s and 102.4s, respectively. Through these experiments, two crucial insights have emerged: 1) Under the same total acquisition time, the results obtained from the 16 * 16 scanning grid are better than those from the 32 * 32 scanning grid, revealing that longer exposure times for fewer scanning points yield better imaging quality. 2) However, when comparing the results from 32 * 32 and 8 * 8 scanning grids, this phenomenon is observed to be less pronounced under extremely low scanning grids.

In light of these insights, we infer that for our specific imaging system, employing a 16*16 scanning grid strikes a balance between the reconstruction quality and the total acquisition time (yet this would vary for different systems). Furthermore, it is noteworthy that our manuscript introduces a universally applicable technique for NLOS imaging utilizing under-scanning measurements. This method is agnostic to the tradeoff scanning grid of any given imaging system, reaffirming its broad applicability.


**More details about our NLOS imaging system**. Our system operates in a confocal manner, with a 532 nm laser emitting pulses at 70 ps pulse width and 20 MHz repetition frequency. The emitted pulses pass through a two-axis raster-scanning galvo mirror, transmitting onto the visible wall. Subsequently, both direct and indirect diffuse photons are collected by another two-axis galvo mirror and coupled to a multimode fiber, directed towards a free-running single-photon avalanche diode (SPAD) detector with a detection efficiency of approximately 40\%. A time-correlated single photon counter records the sync signals from the laser and the photon-detection signals from the SPAD. During data collection, the scanning points and sampling points maintain the same direction but are slightly misaligned to avoid capturing the first bouncing signals during scanning. We employ raster scanning across a 2m * 2m area on the visible wall, and each transient measurement's histogram length is set to 512, with a bin width of 32 ps.

**Fig 2**: More real-world qualitative results.

As suggested by Reviewer iKMu, 9P8E, and VzCp, we extend the evaluation to include more real-world data captured from different NLOS imaging systems. Fig. 2 of the global rebuttal PDF showcases two scenarios: the top scene captured by [1], and the bottom scene captured by our imaging system. As observed, our method delivers promising results with fine details and reduced noise levels. Moreover, it exhibits impressive robustness even under extremely low scanning grids. The demonstrated superiority of our method over other solutions underlines its remarkable generalization capability across diverse real-world scenes.

**Fig 3**: More qualitative results on the Unseen synthetic dataset.

As suggested by Reviewer VzCp, we generate an additional **Unseen** testing dataset by rendering 250 transient measurements from other objects (e.g., baskets, cars, cupboards, etc.). The qualitative results are presented in Fig. 3 of the global rebuttal PDF. As evident from the results, our method outperforms the competition in intensity reconstruction, showcasing the superior generalization capability of our network to unseen objects.

**Table 1**:  Quantitative comparisons on the Unseen dataset.

As suggested by Reviewer VzCp, we generate an additional **Unseen** testing dataset by rendering 250 transient measurements from other objects (e.g., baskets, cars, cupboards, etc.). The quantitative results, including NeTF and CSA, are listed in Table 1 of the global rebuttal PDF. As evident from the results, our method outperforms the competition in intensity reconstruction, showcasing the superior generalization capability of our network to unseen objects.

[1] Yue Li, Jiayong Peng, Juntian Ye, Yueyi Zhang, Feihu Xu, and Zhiwei Xiong. Nlost: Non-line-of-sight imaging with transformer. In Conference on Computer Vision and Pattern Recognition, pages 13313–13322, 2023.

---

### Decision · Program_Chairs · 2023-09-21

**Decision:**

Accept (poster)

**Comment:**

The paper presents a method for non-line-of-sight imaging in the case where a light source and detector are used to illuminate and image a sparse set of points on a relay surface. While the reviews were initially mixed, with concerns about the novelty (i.e., the paper uses a similar reconstruction network to Chen et al. (2020)), evaluation of different scanning resolutions, and lack of baseline comparisons to other methods. The authors addressed these concerns in the rebuttal, with results comparing performance at different resolutions and additional baseline comparisons.

After the discussion phase, all reviewers had a positive rating, and the AC finds the paper suitable for acceptance. While the paper does build on existing deep learning architectures, there is still novelty in applying these methods to a new problem domain. Overall, the paper does a good job at addressing the problem of NLOS imaging with underscanning measurements: the writing is clear, the evaluation is thorough, and the techniques is applied convincingly to real-world datasets. The authors should incorporate the additional resolution analysis and baseline comparisons into the camera ready version. Additionally, the authors may wish to mention and discuss the concurrent work, appearing at CVPR 2023, referenced below.

Wang, Jianyu, et al. "Non-line-of-sight Imaging with Signal Superresolution Network." Proceedings of the IEEE/CVF Conference on Computer Vision and Pattern Recognition. 2023.